# Cognitive Test Solution in Mice with Different Brain Weights after Atomoxetine

Olga V. Perepelkina * and Inga I. Poletaeva *

Biology Department, Lomonossov Moscow State University, Vorobievy Gory, 1, Building 12,
Moscow 119234, Russia
* Correspondence: o_perepel73@mail.ru (O.V.P.); ingapoletaeva@mail.ru (I.I.P.)

**Abstract:** In this paper, the data are presented concerning different reactions to seven daily injections of atomoxetine in two mouse strains differing in relative brain weight. Atomoxetine affected the performance in a puzzle-box cognitive test in a complicated way—the large brain mice were less successful at task solutions (presumably because they were not afraid of the brightly lit test box), while the small brain strain of atomoxetine treated mice solved the task more successfully. The behavior of all atomoxetine treated animals was more active in an aversive situation (an unescapable slippery funnel, (analogous to the Porsolt test) and the time of immobility decreased significantly in all atomoxetine treated mice. The general patterns of behavioral reactions to atomoxetine in the cognitive test and other interstrain differences demonstrated in these experiments made it possible to suggest that differences in ascending noradrenergic projections between the two strains used exist. Further analysis of the noradrenergic system in these strains is needed (and further analysis of the effects of drugs which affect noradrenergic receptors).

**Keywords:** mouse strains; brain weight differences; cognitive task; aversive behavior; atomoxetine; noradrenergic ascending projections





## 1. Introduction

The relative brain weight (brain weight/body weight) is the trait used by evolution biologists as the index of CNS development in species comparisons (e.g., [1–3]). The selection of mice for large and small relative brain weight demonstrated that several differences in behavioral indices are also associated with this trait [4,5].

Three successive selection experiments were performed in our laboratory, with the selection of mice based on large and small brain weight [5].

Large brain mice (LB) demonstrated higher cognitive abilities (in tests for elementary logic task solutions and in learning paradigms) while mice with small brain weight (SB) displayed higher (than LB) anxiety, a tendency for depressive-like behavior and elevated stress-reactivity.

During the third selection experiment, the differences in behavior between LB and SB mice still persisted when the selection for different brain weights stopped at the level of F23 selection generation [5,6]. After five generations of interstrain breeding (without selection) LB vs. SB differences were found not only in brain weight, but in the reactions to alcohol injections, which happened to be similar to those shown in our second selection experiment [6,7]. The interstrain differences, which were persistently found in all three selections, could also include the differences in brain morphology. The data accumulated show that in laboratory mice (which are very popular for neurobiological modelling) brain weight could not only affect the level of cognitive abilities, but could determine several other behavioral peculiarities. It was possible to suggest that LB and SB mice could react differently in response to pharmacological agents.

Atomoxetine (AT), the inhibitor of noradrenaline transporter [8], induces behavioral changes, presumably caused by the increase in brain noradrenaline levels [9–13]. This determined our interest in this drugs' effect in mice, differing by brain weight.

The mice's ability to solve the cognitive test and exploration strategies in a closed plus maze [5] was analyzed, as well as the acoustic startle reaction and the behavior in an "unescapable slippery funnel" (a test similar to the Porsolt test). According to clinical data, the AT effects when used as ADHD treatment are seen not earlier than seven days after treatment [14].

The forebrain structures (neocortex, hippocampus, amygdala, etc.) are innervated by ascending axons from the noradrenergic brain stem nucleus, the locus coeruleus (LC). The LC stimulation affected learning, decision making, attention processes, sleep-wakefulness cycle and sensory processing (pain perception and brain hemodynamics, in particular) [15,16]. The hypothesis, which served as the prerequisite for the present study, was the following: the increase of noradrenaline in the brain could exert different effects on behavioral reactions in LB and SB mice as their noradrenergic networks could be organized in different ways, including ascending projections from the brain stem to forebrain structures. Thus, the analysis of chronic atomoxetine injections on the behavior of animals of LB and SB strains was performed.

## 2. Materials and Methods

Experimental animals. The AT injections were given to 4-month-old, male mice of LB ($n$ = 20) and SB ($n$ = 17) strains, while control animals (LB, $n$ = 23; SB, $n$ = 16) were injected with similar volumes of saline.

The mice were maintained in plastic cages (33 × 22 × 8 cm, 5–6 animals per cage) with mouse chaw (Laboratorkorm) and fresh water ad lib.

Atomoxetine injections. Atomoxetine (AT, Straterra, firm Eli Lilly Pharma, Madrid, Spain) was injected i.p., over 7 days, in doses 2 mg/kg (for F13 mice) and 2.5 mg/kg (for F14 mice). After the course of injections, the animal behavior was analyzed during 3 successive days (see below).

The "puzzle-box" cognitive test [17]. A shorter protocol version of the "puzzle-box" cognitive test was used, based on defensive behavior—the urge to hide, avoiding a brightly lit environment (Figure 1A). The test estimates the animal's ability to "understand" the object permanence rule (an object, perceived recently but not seen any more, still exists and can be found). The experimental plastic box had two compartments, one brightly lit (30 × 28 × 27.7 cm) and one dark (14 × 28 × 27.5 cm), which were connected by an underpass inserted below the floor surface (4.5 × 11.5 × 1.5 cm). An animal introduced into the light part of the box is eager to avoid the unpleasant surrounding and enters the dark part of it. The test procedure contained 4 stages: in the 1st stage, the underpass was unobstructed and an animal could enter the dark part for free; in the 2nd stage, the underpass was masked by fresh wood shavings at the level of the box floor and an animal could penetrate the dark environment by removing this obstacle with its paws; and in the 3rd and 4th stages, the underpass was blocked by a light (carton-plastic) plug, which the mouse could remove with its paws and teeth. Success was evaluated by the length of time taken to enter the dark compartment. Animals were given 180 s (for stages 1 and 2) and 240 s (for stages 3 and 4) to perform the task. If a mouse did not perform the solution in the time indicated, the behavior was ranged as "non-solution". In these experiments, an important index was also the latency of the first approach to the underpass area after the mouse was placed into the lit compartment. The proportion of animals in a group that was able to solve each of the 4 task stages was also an important index of the group's ability to solve the task.

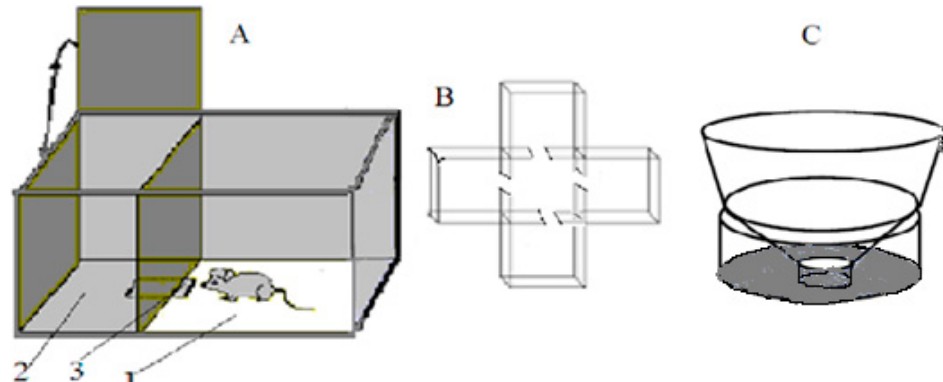

**Figure 1.** Schematic drawing of (**A**) the puzzle-box experimental box featuring (1) brightly lit part of the puzzle-box, (2) the dark part of the box and (3) the underpass, connecting both parts; (**B**) the closed plus-maze; and (**C**) an unescapable slippery funnel.

Closed plus-maze test (CPM). A CPM was performed in a special plastic box (Figure 1B). The apparatus consisted of 4 closed empty arms (15 × 15 × 15 cm) connected to a similar sized central compartment via opened doorways (7 × 7 cm). The animal was placed into the central compartment and allowed to explore the maze. The sequence and timing of arm visits were registered into computer until 13 visits had been made. The criterion for a visit was the animal's entry into a compartment with all four paws. Subsequent computer analysis was made to reveal several independent behavioral measures [18]. The processing program of animal behavior in CPM [19] allowed the recording of several behavioral reactions, partly revealing the interstrain or intergroup variability. These were: the time spent by an animal in maze arms (indicating the exploration tendency); the number of visits to arms before the mouse visited all four arms ("patrolling" index, being connected to acquisition of spatial information); and the latency of the first visit to an arm, indicating animal anxiety in the novel (although not frightening) environment. The tendency to alternate visits into 2 (out of 4) arms, as "stereotype behavior" expression, could be evaluated by the program as well.

Acoustic startle reaction. The acoustic startle reaction was registered in response to a sound click (9 k-hertz, 108 dB, time curve—0.2 s). The click was given 5 times with variable (no less than 15 s) intervals between them. The click was introduced when an animal had all 4 paws on the floor. The arbitrary units of reaction intensity were: 0—no visible reaction; 1—freezing; 2—jerk; and 3—jump [20].

"Unescapable slippery funnel" test (Figure 1C). This test is regarded as analogous to the Porsolt unescapable swim test. In contrast to the latter, it does not require the full immersion of an animal in water, thus preventing possible artefacts due to different amounts of fat deposits in mice of different groups. The device consisted of an upper cylindrical portion with vertical walls (8 cm, diameter 27.5 cm), the "funnel section" (9.5 cm high) with walls sloping outward (angle of 40° from horizontal) and a lower cylindrical funnel "gorge" with vertical walls (4 cm high, 4.5 cm in diameter). Water (approximately 22 °C) filled the bottom cylinder and its depth was such that, when standing immobile at the bottom, the hind legs (up to femur) of an animal were immersed in the water. During the 3 min test, the observer recorded (using a computer program) the time spent in three alternative states: (1) immobility in the water at the bottom of the funnel; (2) climbing out of the water acquiring a sprawling posture with all four paws out of the water (avoidance); and (3) attempts to climb or jump out of the funnel (active escape).

The elevated plus maze (EPM) test. The EPM test was given to mice of F14 only. The plus-shaped part of the device was elevated (40 cm) above the surface, with the length of arms—25 cm and arm width—5 cm. Test duration was 3 min.

Statistics. The scores of latency to enter the dark compartment of the puzzle-box, latency of the first approach to the underpass and the behavioral scores of CPM, EPM, startle reaction and slippery funnel behavior were evaluated for significant differences by means of one- and

two-factor ANOVA (factors "strain" and "treatment") with subsequent post hoc Fisher LSD analysis. The significance in the proportion of animals that solved the "plug" stages of the puzzle-box test were evaluated using φ Fisher criterion for alternative proportions.

## 3. Results

As mentioned above, LB and SB mouse strains are now maintained without selection for high and low relative brain weight using the random mating inside each of these strains (the selection procedure was not performed after F24 of our third selection) experiment [5]. Nevertheless, in animals of F13 and F14 (bred without selection, F42–43 from the start of the third selection) the absolute brain weight and brain/body weight ratios were significantly different (Figure 2). The differences in body weights were non-significant (27.7± 0.5 g and 26.6 ± 0.5 g, for LB and SB respectively).

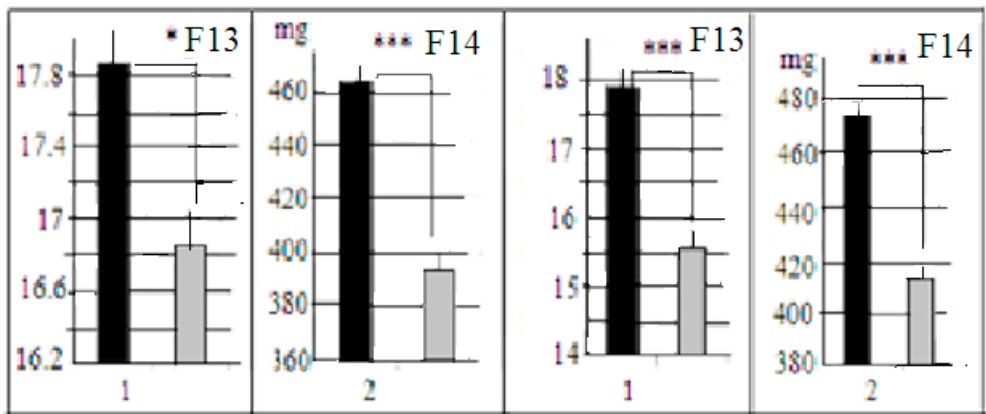

**Figure 2.** Mean values (±st. err., ordinate) of (1) brain weight, mgs/body weight gs ratios and of (2) absolute brain weights for all mice from F13 and F14. Designations: black columns—LB, grey columns—SB mice. *, *** significantly different (1-factor ANOVA, factor "strain", LSD post hoc by Fisher).

Puzzle-box behavior data (in spite of slight differences in AT doses used) were obtained for mice of two generations. They demonstrated similar patterns of interstrain differences and of AT effects. No statistically significant differences between the scores of experiments for two generations were found and data are presented summarily.

Puzzle-box test. As mentioned in methods section, one of the indicators for success in the puzzle-box was the proportion of animals which solved the task, as well as the entrance latency into the darkness.

Differences in the time taken to solve the open underpass stage were significant for control groups only, with more quick solutions in LB mice (Figure 3I, Figure 3II), indicating interstrain differences. It should be noted that in SB mice the latencies for AT and control groups were more or less of similar values, while in LBs there was the tendency ($p = 0.1$) for longer latencies in AT groups, i.e., more slow reactions of LB mice after AT (at the stage of open underpass). The similar pattern of differences was for stage 2, when the underpass was masked by wood shavings (Figure 3II).

The data on intergroup differences in latencies for task solution for stages 3 and 4, when the underpass was blocked by the plug, are not very informative, as in all groups there were animals which failed to solve these task stages and their latencies were assumed to be 240 s, thus affecting the mean values of latencies. Although the 2-factor ANOVA for 1st "plug" trial latencies reveal the significant influence of "strain" factor with more quick task solution by LB mice ($F_{1-2} = 4.028$, $p = 0.048$).

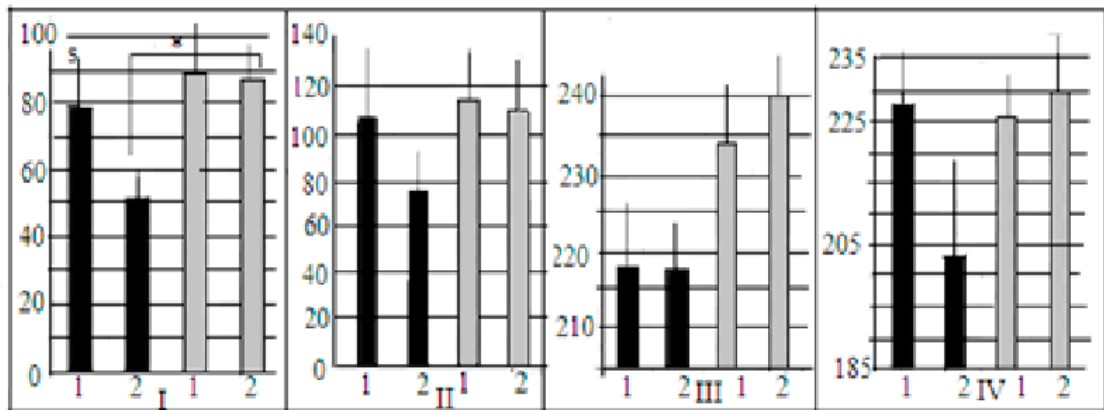

**Figure 3.** Mean latencies (s, ordinate ± stand. err) of mice entrances into dark compartment of the experimental box in the course of successive puzzle-box stages by LB and SB mice ((**I**)—the underpass is unobstructed, (**II**)—it was masked by wood shavings, (**III,IV**)—it was blocked by a plug). Designations: black columns—LB mice; and grey columns—SB mice. 1—mice groups after AT injections. 2—control groups (saline). *—significant differences between control groups (2-factor ANOVA, factors "strain" and "treatment", LSD post hoc criterion by Fisher).

In both generations of mice used in our experiments, the interstrain differences between control and AT groups of LB and SB could be better illustrated by the proportion of animals that solved the puzzle-box stage with the plug (those animals who were able to penetrate into the darkness due to "goal-directed" plug removal) (Figure 4). The 2-factor ANOVA (factors "strain" and "treatment") revealed the significant influence of "strain" factor ($F_{1-2} = -27.75$, $p = 0.0000$), as well as the significant "factor interaction" influence ($F_{1-2} = 36.31$, $p = 0.00000$), which denoted the opposite "direction" of differences control vs. AT groups in LB and SB.

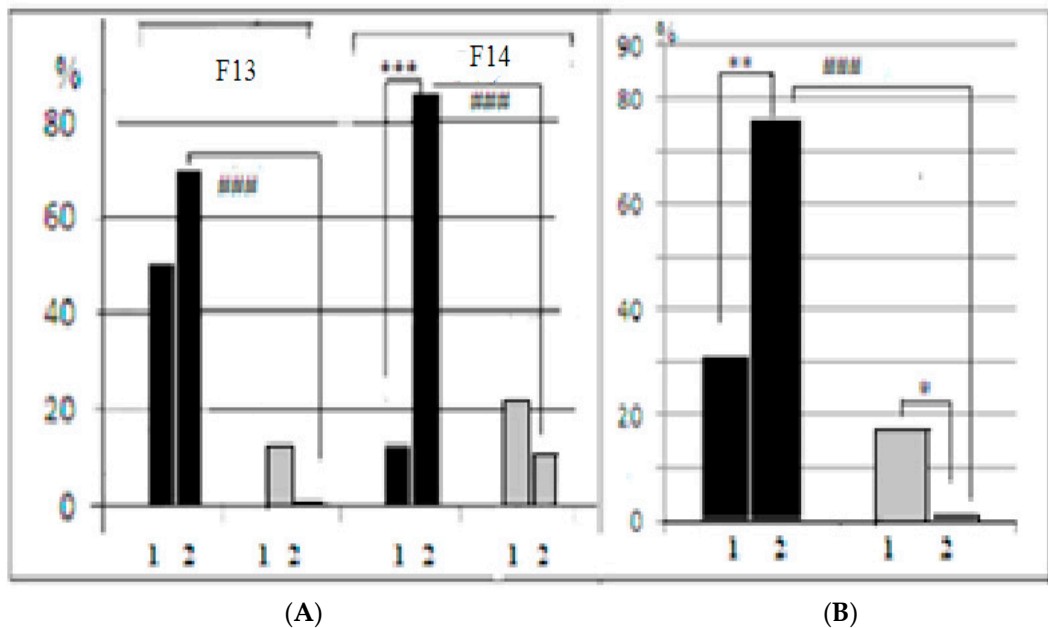

(**A**)  (**B**)

**Figure 4.** (**A**) Proportion (%, ordinate) of LB and SB mice which were able to solve the puzzle-box stages when the underpass was blocked by the plug in F13 and F14. (**B**) Summarized scores for two generations. Designations are as in Figure 3. *, **, ***—significant LB and SB intergroup differences (AT vs. control) ($p < 0.05$, 0.01 and 0.001, respectively), ###—significant differences between LB and SB control groups, $p < 0.001$ (φ criterion for alternative groups).

In LB mice of F14 (Figure 4A), the proportion of animals that solved the first plug stage after AT treatment was significantly ($p < 0.001$) lower in comparison to the control group, while in F13 this difference was similar by sign, but not significant. The summarized scores for both generations demonstrated the significance of AT vs. control difference (Figure 4B). In contrast to the LB data, in the SB mice AT group several animals solved the task at this stage, while in the SB control groups no mice at all solved this task stage. This resulted in a significant difference AT vs. control ($p < 0.05$). The results of the test stage 4 (the second "plug" stage) for SB were practically similar to those for the first "plug" stage, while in the LB group, after the AT, the proportion of task solutions was non-significantly higher than in the "first plug" and in the AT group of SBs. The respective comparisons of scores for the control groups demonstrated the significant prevalence of the "plug" stage solutions in LB mice, confirming the presence of interstrain differences in this cognitive test solution. It is also informative to note that latencies of animal first (investigative) approach to the underpass were similar in two LB groups while this latency was significantly longer in SB group after AT. The influence of the "strain" factor for this behavioral index, according to 2-factor ANOVA, was significant ($F_{1-2} = 14.36$, $p = 0.0003$)—the latencies for the first underpass approach (for both AT and control animals) in SB mice were higher than those for LB. The influence of the "treatment" factor was also significant ($F_{1-2} = 7.99$, $p = 0.00$), as well as the "factors interaction" ($F_{1-2} = 6.23$, $p = 0.015$). The latter reflected the fact that in SBs AT group the latency of the first approach was higher than in control group, which was not the case in the LB strain.

The expression of fear (in an unpleasant novel environment) in rodents and laboratory mice could be evaluated by measuring freezing behavior. Thus, the number of freezing episodes (immobile mouse posture with no vibrissae movement) could indicate the degree of fear experienced by an animal. The mean number of freezing episodes (summarized for all 4 test stages) in the LB group after AT (Figure 5B) was significantly higher than in the control group, while in the SB groups the effect was opposite—the mean number of freezing episodes in AT group was lower than in the control. It is interesting that the absolute numbers of freezing episodes in the AT groups of both strains were equal (Figure 5B).

The defecation values, also estimated for 4 test stages in sum, were significantly lower in the AT groups of both strains in comparison to the control (Figure 5C). According to the traditional logical explanation, the defecation level is the index of emotional reactivity (i.e., the fright of an animal). The difference between the AT group and the control indicates the lower fright of animals in the novel environment after AT.

In the closed plus-maze test (CPM), the pattern of AT influence on maze exploration in LB and SB mice was also rather complicated. In general, according to present data and information on LB and SB strains obtained earlier [5], the LB mice explored this maze more actively, demonstrating the "goal-directedness" in their behavior. AT treated LB mice displayed significantly more "patrolling" cycles (see Methods) than AT treated SB mice (LB—$2.12 \pm 0.26$, SB—$4.0 \pm 0.26$, $p = 0.047$), which means that LBs visited maze arms more "methodically", and this fact could serve as the argument against the fear-inducing AT effect in the puzzle-box (see Section 4). The number of "steps ahead", i.e., visits to the arm other than the one entered previously is also regarded as the expression of exploration behavior. In LB mice after AT these "steps ahead" were significantly more numerous than in SB after AT ($p = 0.003$), and more numerous than in LB control ($p = 0.018$). LB mice spent significantly more time in maze arms (in the course of two "patrolling" cycles) than SBs, the data for AT groups being significant (LB, $21.9 \pm 4.4$ and SB, $4.0 \pm 4.1$, $p = 0.005$). The time spent in the center before the start of maze exploration (indicating animal "eagerness" or "reluctance" for exploring the novel place), was shorter in both AT groups in comparison to the control (data not presented).

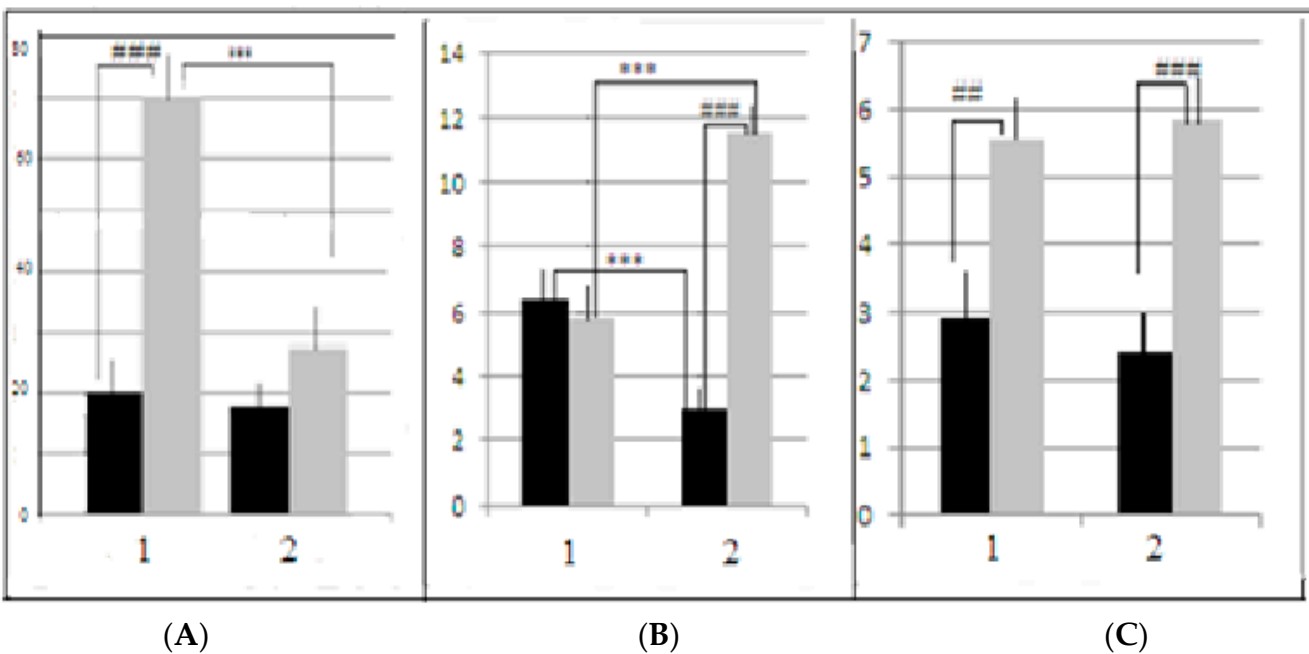

**Figure 5.** (**A**) Mean latencies (ordinate, s ± stand. err.) of the first approach to the underpass. (**B**) Mean numbers of freezing episodes (ordinate, ± stand. err.), summarized for 4 puzzle-box test stages in LB and SB groups, AT (1) and control (2) groups. (**C**) Summarized defecation scores, Designations are as in Figure 3. \*\*\*—significant differences between AT and control groups, *p* < 0.00; ##, ###—significant differences between the LB and SB groups, for *p* < 0.01 and *p* < 0.001, respectively, (2-factor ANOVA, "strain" and "treatment as factors, LSD post by Fisher). black columns—LB mice, grey columns—SB mice.

The differences in the acoustic startle are as follows: in response to the delivery of clicks, from 1st to 4th, the startle reaction in AT treated LB mice was significantly more intense than in the control LB group; and in both SB groups the p value varied from 0.018 to 0.003. The 5th click delivery induced startle of higher intensity in the LB group after AT, in comparison to two other groups, but the difference was not significant. It was shown previously that AT enhanced the prepulse inhibition in mice [21]. The startle reaction intensity in both groups of SB mice did not differ between AT and control groups. It was shown earlier [22], that 20 mg/kg (but not 0.2 mg/kg) of AT increased the startle reaction in outbred mice.

As already mentioned, the "sign" of differences between the AT and the control groups in both strains was similar to behavioral indices in the unescapable slippery funnel (the data for LB and SB being summarized). The mean time of immobility was shorter in mice after AT treatment (Figure 6). Although in spite of the fact that differences between the AT and control groups were of similar "sign"' in both strains, LB mice were, as a whole, more active than SB, which was demonstrated by the 2-factor ANOVA results, factor "strain" demonstrating the significant influence, ($F_{1-2}$ = 7.5102, *p* = 0.0080). The time occupied by active reactions during 3 min test in LB mice after AT was longer than in SB (tendency, *p* = 0.076), while in the SB control it was significantly longer than in respective group of LB (*p* = 0.0424). Thus the behavior of the AT treated mice in the aversive situation (part of body immersed in water, see Methods) was more active while the time, they spent in immobility—shorter. In other words, in a situation when the "cognitive" behavioral component was not decisive for the current behavioral adaptation, the AT effect was expressed as behavior activation (which is also the argument against the idea of fear-inducing effect of AT in puzzle-box).

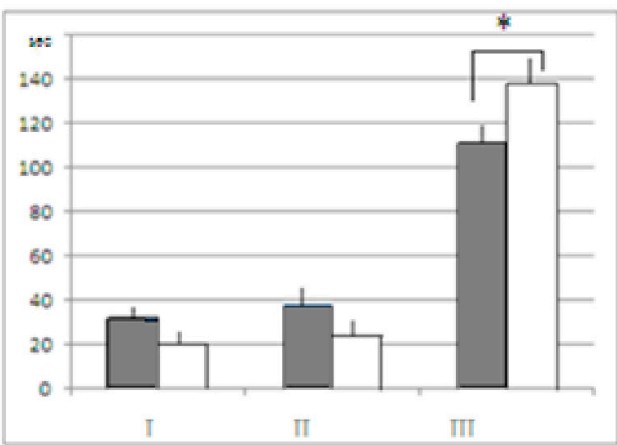

**Figure 6.** Mean time (ordinate, s, ± stand. err.) spent by mice of both strains in the unescapable slippery funnel. I—active jumps, II—stretched posture of an animal in the funnel gorge above the water surface, III—immobility. White columns—control groups, dark grey columns—mice after AT treatment. *—significant difference (1-factor ANOVA, $F_{1-2}$ = 5.6218, $p$ = 0.020761).

The results of the EPM test also showed the similar pattern of differences between the AT and control groups for LB and SB. The time spent in open EPM arms was shorter in mice after AT. The 2-factor ANOVA showed the significant influence of "strain" ($F_{1-2}$ = 14.51, $p$ = 0.00046) and significant influence of factor "treatment" ($F_{1-2}$ = 3.4, $p$ = 0.043). The shorter time spent in open arms by animals of both AT groups indicated their elevated anxiety (or caution?). According to the Fisher post hoc LSD test, the intergroup differences were at the level of "tendency" for LB mice ($p$ = 0.07) and were not significant in SB ($p$ = 0.44). The difference in the numbers of freezing episodes in EPM did not coincide with the "sign" or with the respective data from puzzle-box test. According to the 2-factor ANOVA, there was the significant influence of "strain" factor ($F_{1-2}$ = 6.13, $p$ = 0.016), as well as significant "factor interaction" ($F_{1-2}$ = 8.28, $p$ = 0.0 009). In the LB mice after AT, the EPM freezing episodes were less numerous than in the control (3.0 ± 0.94 vs. 5.43 ± 1.06, $p$ < 0.09, tendency), while in the SB group after AT, freezing episodes were more numerous (although not significantly) than in the control (9.0 ± 0.94 vs. 7.71 ± 1.06).

## 4. Discussion

Summarizing the data for the puzzle-box test, it is clear that interstrain differences in AT groups of LB and SB were of the opposite sign in comparison to control groups. It is possible, that longer escape latencies in LBs after AT during stages 1 and 2 of the puzzle-box test and the lower proportion of animals that solved "plug" stages could be due to LB strain-specific characteristics. This specificity could be explained in two opposite ways. The first is that AT injections resulted in the increase in fear in LB mice as the response to being placed into the novel environment (brightly lit place). The fear reaction of this category depends on axonal projections on *colliculi superior* neurons from LC [23]. These facts could mean that the fear experienced by an animal could affect the adequacy of behavior by inhibiting novel place exploration and inducing the longer reaction times. So it could happen that increased fear after AT influences LB mice to move more slowly and to solve the most "difficult" test stage less efficiently. Another possible explanation of the decrease in the puzzle-box scores in LB after AT is the opposite one. It could be that mice of LB strain after AT (but not SBs) experienced less fear in the new environment, and were reluctant to escape in the darkness. In other words, their fear motivation, which normally triggers the escape into darkness, decreased as the result of the AT effect. This suggestion is supported by the lack of differences in the latencies of the first approach to the underpass between two LB groups (Figure 4A), while this latency was significantly longer in SB group after AT. So it could be suggested that AT-mice of the LB strain were less afraid to approach the

underpass, in contrast to the similar SB group in which the latency of the first approach was longer than in respective control group.

As already mentioned, in the AT group of SB strain were mice that solved the "plug" stage, while no animals solved it in SB control group. It is possible to suggest that AT injections in the SB decreased the cautiousness of these animals, making it possible for part of them to solve this test stage.

The complicated pattern of AT influence on anxiety reflects the plausible role of individual variability, as both strains (LB and SB) are outbred. One more important issue is that the anxiety experienced by an animal (both "trait" and "state" anxiety) is not the unitary physiological state of the brain. Anxiety reactions reveal the complicated neurophysiological basis and different expression in different test environments [24–30]. The fact that traditional indicators of anxiety and their changes in response to AT were not similar in EPM and puzzle-box in LB and SB mice is not surprising. It also means that the AT effects on the anxiety index in different tests in LB and SB mice is open for further experimental analysis. In effect these discrepancies are worth analyzing with a special investigation using the battery of "anxiety" provoking tests. In the rather simple EPM test (in which a mouse experiences anxiety as well as the certain urge to explore the open arms) the AT injected SB mice had more freezing episodes than the control group, while in the puzzle-box they froze less. Our data shows that the influence of AT was different for LB and SB in these two tests. The analysis of the differences in the tests used does not permit us to state that the AT injections increased anxiety in LB and decreased it in SB (while data on the freezing episode numbers is the "pro" argument for such conclusion). In any case, it is more or less clear that AT exerted not similar changes in the behavior of LB and SB mice in two different experimental paradigms—in a cognitive test and in EPM. It is worth mentioning that the change in brain noradrenergic function (KO for alpha-1b adrenergic receptor) resulted in the increase in reaction to novelty, reducing learning capacity at the same time [31]. This study illustrates the subtle "tuning" of brain processes in norm and demonstrates the difficulties in interpreting data when "overall" changes in the brain noradrenergic system occur. The atomoxetine exerts rather noticeable effects on behavior traits, differing in their function, which could be due to certain misbalances in the multifaceted brain system. The data on the AT effect for scores of the EPM test probably reveal the high individual variability in mice groups in EPM anxiety test, which coincides with the high individual variability demonstrated for outbred mice [32,33].

As mentioned above, noradrenergic activation influences behavior in tasks which require attention, as well as affecting the general "arousal", learning and memory. It is important to underline that noradrenergic neurons are present not only in LC, but in the A2 neuronal group (the so called dorsal vagal complex), which is situated in dorsomedial part of the medulla [34]. The A2 neuronal group is involved in the display of sensory-motor and visceral reactions, and this group is known to project (reciprocally) into brain stem nuclei (including LC), hypothalamus and limbic system. L. Rinaman [34] noted that most investigations, summarizing the role of ascending noradrenergic projections, addressed the LC influences only, while the A2 neuronal group function was ignored. The A2 projections into hypothalamus and in several forebrain structures were described, which provide CNS reactions in response to stimulation. This information is important as it may indicate the plausible reasons of interstrain differences in AT infusion, demonstrated here.

According to the pattern of puzzle-box freezing episodes, the AT vs. control group differences in LB and SB mice were opposite, while "emotional reactivity" differences AT vs. control were similar for both strains. It is possible to suggest, that this "discrepancy" is due to interstrain differences in noradrenergic LC and A2 neuronal projections [34,35]. It could also mean that the common neurophysiological interpretation of interstrain differences in cognitive abilities and anxiety indices is not "working" in the case when the brain noradrenaline level was increased artificially (due to AT application). This issue could not be discussed here in detail as the data presented are not sufficient for such comparisons. The causes of behavioral differences, induced by this pharmacological intervention, should

be researched with the peculiarities of noradrenergic axons in the forebrain structures. It was demonstrated earlier [22], that AT infusions (3 mg/kg) increased attention scores only in mice with the initial low attentional level, and the same pattern was found in the murine KO model of decreased attention (i.e., KO vs. wild type animals) [31]. These data are in parallel with different interstrain changes after AT treatment. It is also interesting that heterozygous KO mice for noradrenergic transporter (with increased brain noradrenaline levels) showed higher anxiety level in a light-dark test than in wildtype animals [36]. This test paradigm is more or less similar to the puzzle-box stage 1, with an open underpass. The genotype dependent higher scores of the freezing reaction were found in the conditioned fear paradigm after AT treatment as well [37]. Thus, the pattern of behavioral changes in mice of different genotypes permits us to suggest the presence of genotype differences in the organization of noradrenergic forebrain projections.

It is also possible that behavioral changes induced by AT in an aversive situation (slippery funnel test), which were similar in both strains, could be the result of A2 neuronal system activation, while the peculiar pattern of interstrain differences in cognitive task and CPM exploration could be due to altered noradrenergic signaling from LC. The different "sign" of AT-effects was shown earlier in LC phasic and tonic neuronal reactions in response to sensory stimulation [9]. These data could be the indirect confirmation of such a suggestion (the A2 neuronal activity was not analyzed in [9]).

It is possible to suggest that the "cognitive component" of the brain reaction to AT injections, which is connected to the increase of neocortical noradrenaline (and prefrontal cortex) [37] is not expressed in all cases. The increase of noradrenaline, as the effect of AT, could occur in brain regions which receive the ascending projections (according to [34]) from the A2 area. The complicated pattern of changes in brain structure function after AT could be illustrated by the effect of this drug on histamine levels [38,39], as well as by the specific pattern of neocortical noradrenergic projections [40].

The LC functional participation and detailed morphological issues, which are now established in developmental, genetic, molecular, anatomical and neurophysiological studies, requires also admitting that the LC participation in brain function is to be reconsidered (see [16]). In particular, the functional role of neuronal activation in cellular units, which are known to have the vast projections, raises inevitably the issue of their collateral distribution, which at present is rather difficult to visualize (although could be crucial for understanding the nuances of noradrenergic function). The range of difficulties of this sort could be illustrated by a study, in which the LC projections to the prefrontal and primary motor cortex were investigated by means of complicated methodology [41].

The analysis of the role, which genetic peculiarities play in norm and after pharmacological interventions, is important for perfection of genetic and/or pharmacologic models of human CNS diseases [42]. At the same time, such knowledge is also indispensable for the study of the fundamental scientific problem, that of the role which definite behavioral trait and brain weight peculiarities play in cases of high or low animal cognitive abilities levels. Of course, the suggestions described above need to be confirmed by both- pharmacological analysis using drugs specifically addressed to the noradrenaline receptors (as in [43]), and immunohistochemical investigations of brain structures in respect to noradrenaline receptors expression in several brain areas, using rodent strains which differ in AT injections effects.

Concluding the data presented, it is important to note that it is the first evidence (and unavoidably preliminary one) of AT effects in mouse strains, which differ in the relative brain weight scores. This inevitably means that further experiments are needed for more detailed analysis of effects described, in particular with more experimental details concerning animal anxiety changes as the result of such "intervention" in brain noradrenergic system balance. As mentioned above, the indices of the morphology and/or immunohistochemistry of noradrenergic system in these strains could make the issue more interesting in general and more convincing (than the behavioral data only). This is also true for any other behavioral and neurochemical genetic comparison when the noradrenergic system is

involved, including data on mice with altered noradrenergic receptors (which we were not able to analyze here).

**Author Contributions:** O.V.P. and I.I.P. participated equally in performing experiments, O.V.P. performed the breeding of mice and the determination of brain weights in two generations of LB and SB mice. I.I.P. wrote the text. All authors have read and agreed to the published version of the manuscript.

**Funding:** Supported by Russian Scientific Foundation, grant 23-25-00042.

**Institutional Review Board Statement:** Bioethics for LB and SB mouse strains breeding and maintenance were confirmed by Lomonossov Moscow State University Bioethical Committee (№ 49, from 18 June 2014).

**Informed Consent Statement:** Not applicable.

**Data Availability Statement:** Not applicable.

**Conflicts of Interest:** The authors declare no conflict of interest.

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
