# Peer review of "Cognitive Test Solution in Mice with Different Brain Weights after Atomoxetine"

_2035-8377, doi:10.3390/neurolint15020041_

Round 1

Reviewer 1 Report

The article is hard to read. Authors should write a hypothesis or research question at the end of the introduction. Instead, they summarise their findings. They also mixed results and discussion section. I would advise they remove all explanation from the results and move it to the discussion. In the discussion, I miss the critical evaluation of their results. As it is written now, their study has no limitations or shortcomings.

Minor. They have to check for typos and improve their English.

Author Response

We wish to express our gratitude for the efforts the estimated reviewers made reading and evaluating our paper.

We will answer the questions and remarks in more detail below, the changes in the text are marked by yellow colour.

You wrote: “Minor. They have to check for typos and improve their English”

Our reply: Typos discovered were corrected. We also did our best in making the English better.

You wrote: Authors should write a hypothesis or research question at the end of the introduction.

Our reply. Thank you for the remark! The respective phrase now  looks as: “The hypothesis which served as the prerequisite for the present study was the following. The increase of noradrenaline in the brain could exert differential effect on behaviour reactions in LB and SB mice as their noradrenergic networks could be organized in different way including the ascending projections from brain stem to forebrain structures especially. Thus, the analysis of chronic atomoxetine injections on behaviour of animals of LB and SB strains was performed”.

You wrote: They also mixed results and discussion section. I would advise they remove all explanation from the results and move it to the discussion. In the discussion, I miss the critical evaluation of their results. As it is written now, their study has no limitations or shortcomings.

Our reply. Thank you very much for this comment (of course we are aware of limitations and shortcomings of these data)! And we introduced the respective phrases in the text. We switched most part of the discussion in the Discussion section.  As for critical evaluation of results, we mentioned not once that these data are the first in the respect of genetic (brain weight) differences in reaction to atomoxetine, and we noted as well, that the morphology and immunohistochemical data are badly needed in order to visualise the problem in more broad sense. The same is true for more detailed analysis of the anxiety reactions.

Reviewer 2 Report

The manuscript “Cognitive test solution in mice with different brain weight after atomoxetine” by Perepelkina et al is a research article which examined the effects of atomoxetine, a noradrenaline uptake inhibitor, on the cognitive functions in two mouse strains differing in relative brain weight. The authors found that atomoxetine differentially affected the solution of puzzle-box cognitive test in the two strains. In conclusion, the authors suggest that these differential effects of atomoxetine are probably due to differences in ascending noradrenergic projections between the two strains. Generally, the subject is of interest and important in its field. Also, this work is scientifically sound and contains essential contents. This paper is also of importance for providing us the important evidence that differences in ascending noradrenergic projections would exist in two strains (large and small brain weight strains). However, I have some concerns on the paper.

1. In bar graphs, all the data plots should be displayed if possible. The readers can obtain more information from these data.

2. Have the authors tested agonists or antagonists of noradrenaline receptor in addition to atomoxetine? These data would strengthen the conclusion of the manuscript. Please discuss this point!

Author Response

We wish to express our gratitude for the efforts the estimated reviewers made reading and evaluating our paper.

We will answer the questions and remarks in more detail below, the changes in the text are marked by yellow colour.

You wrote:  In bar graphs, all the data plots should be displayed if possible. The readers can obtain more information from these data.

Our reply: Sorry, we did not catch what you mean -  the data plot to be presented numerically?

You wrote: Have the authors tested agonists or antagonists of noradrenaline receptor in addition to atomoxetine? These data would strengthen the conclusion of the manuscript. Please discuss this point!

Our reply: We have not tested noradrenaline receptors agonists and antagonists, because this work started recently and because of technical reasons. Our animal facilities are not “rich”, and we have the limited numbers of animals for experiment (as the breeding process is also on line). So we have not many animals of each strain for experiments. The idea of more thorough analysis of the brain noradrenaline network is expressed in the text now! Thank you!

Round 2

Reviewer 1 Report

Authors improved their work.

 Minor editing of English language required.